# REX: GPU-Accelerated Sim2Real Framework with Delay and Dynamics Estimation

**Bas van der Heijden**                                              *b.heijden@hotmail.com*
*Cognitive Robotics*
*Delft University of Technology*

**Jens Kober**                                                        *j.kober@tudelft.nl*
*Cognitive Robotics*
*Delft University of Technology*

**Robert Babuska**                                                    *r.babuska@tudelft.nl*
*Cognitive Robotics, Delft University of Technology,*
*CIIRC, Czech Technical University in Prague*

**Laura Ferranti**                                                    *l.ferranti@tudelft.nl*
*Cognitive Robotics*
*Delft University of Technology*

**Reviewed on OpenReview:** *https://openreview.net/forum?id=O4CQ5AM5yP*

## Abstract

Sim2real, the transfer of control policies from simulation to the real world, is crucial for efficiently solving robotic tasks without the risks associated with real-world learning. However, discrepancies between simulated and real environments, especially due to unmodeled dynamics and latencies, significantly impact the performance of these transferred policies. In this paper, we address the challenges of sim2real transfer caused by latency and asynchronous dynamics in real-world robotic systems. Our approach involves developing a novel framework, REX (Robotic Environments with jaX), that uses a graph-based simulation model to incorporate latency effects while optimizing for parallelization on accelerator hardware. Our framework simulates the asynchronous, hierarchical nature of real-world systems, while simultaneously estimating system dynamics and delays from real-world data and implementing delay compensation strategies to minimize the sim2real gap. We validate our approach on two real-world systems, demonstrating its effectiveness in improving sim2real performance by accurately modeling both system dynamics and delays. Our results show that the proposed framework supports both accelerated simulation and real-time processing, making it valuable for robot learning.

## 1 Introduction

Sim2real, the transfer of control policies from simulation to the real world, is crucial in robotics thanks to its ability to solve tasks efficiently without the risks associated with real-world learning (Tan et al., 2018; Rudin et al., 2022). With recent advancements in physics simulation on accelerator hardware (Todorov et al., 2012; NVIDIA, 2020; Hu et al., 2020; Freeman et al., 2021), parallelized simulations have greatly reduced training times for complex tasks (Rudin et al., 2022; Hoeller et al., 2024). However, discrepancies between simulation and reality, such as unmodeled dynamics, often reduce the effectiveness of these policies in real-world applications. Addressing this 'sim2real' gap is essential for effective transfer of policies from simulation to the real world.

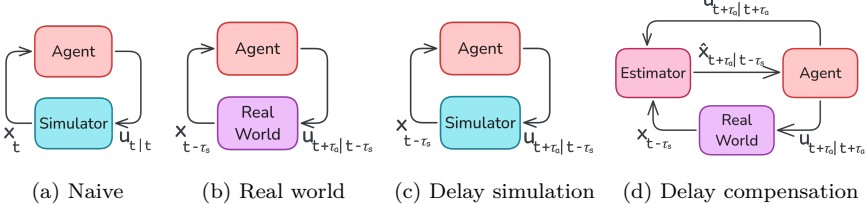

(a) Naive      (b) Real world      (c) Delay simulation      (d) Delay compensation

Figure 1: A policy trained in simulation (a) may perform suboptimally in the real world (b) due to delays. The notation $u_{t+\tau_a|t-\tau_s}$ denotes that an action $u$ is applied at $t + \tau_a$ based on information up to $t - \tau_s$, where $\tau_a$ and $\tau_s$ are the actuation and sensing delays, respectively. By simulating these delays during training (c), the sim2real gap with (b) can be reduced. Alternatively, an estimator (d) can predict future states and compensate for delays, improving policy transfer from (a) to (d). The notation $x_{t+\tau_a|t-\tau_s}$ denotes that a state $x$ is predicted at $t + \tau_a$ based on information up to $t - \tau_s$.

A critical yet often overlooked issue in sim2real transfer is the impact of latency in real-world systems, which can degrade performance (Tan et al., 2018; Ibarz et al., 2021; Elocla et al., 2023; van der Heijden et al., 2024b). The real world is inherently asynchronous, with delayed sensor data causing agents to act on outdated information. Additionally, slow policy evaluations can further delay the agent's actions, compounding these latency issues and leading to suboptimal performance. To mitigate these effects, Fig. 1 illustrates two common compensation strategies: simulating delays during training (Fig. 1c) (Tan et al., 2018; Elocla et al., 2023; van der Heijden et al., 2024a) and using an estimator to predict future states (Fig. 1d) (Smith, 1957; Sherback et al., 2006; Augugliaro et al., 2014; Yang et al., 2020). However, both strategies have limitations. Delay simulation complicates training because the agent's input must include a history of observations and actions to restore the Markov property, while an estimator requires accurately modeled system dynamics and delays, which are often difficult to identify (Unbehauen & Rao, 1990).

The hierarchical and asynchronous nature of robotic systems further complicates accurate and efficient simulation on accelerator hardware. Unlike conventional RL, which assumes a single, synchronized environment (Brockman et al., 2016), robotic systems consist of interconnected models operating at different rates, with asynchronous communication introducing complexities like inter-model latencies and stochastic dynamics (Quigley et al., 2009; Baheti & Gill, 2011), leading to irregular computation patterns. Irregular execution paths require serialization, reducing GPU efficiency, and while simulating time-scale differences improves sim2real accuracy, it further exacerbates this inefficiency (Shibata, 2010).

The main contribution of this paper is a sim2real framework, REX (Robotic Environments with jaX), that introduces a graph-based simulation model with latency effects, optimized for parallelization on accelerator hardware. The framework's innovation lies in its ability to simulate asynchronous, hierarchical systems by explicitly modeling computation, communication, actuation, and sensing delays, while incorporating delay compensation strategies for improved sim2real transfer. Parallelization in both state and parameters allows for simultaneous estimation of system dynamics and delays from real-world data, efficiently minimizing the sim2real gap. Additionally, it supports real-world deployment by distributing computations across CPU cores and accelerators, optimizing for latency and performance.

For RL and robotics practitioners, this framework offers several advantages. It enables the modeling of both simulated and real-world systems through a unified, ROS-like graph-based pipeline (Quigley et al., 2009). The framework supports accelerated training speeds familiar to RL workflows and reduces the sim2real gap by refining models with real-world data. Integration with the JAX (Frostig et al., 2018) ecosystem further supports advanced RL training and optimization (Lange, 2022b;a; Tang et al., 2022; Lu et al., 2022).

Building on these advantages, we make four key claims: Our framework (i) enables the identification of both dynamics and delays from real-world data, (ii) implements delay compensation and simulation techniques that are essential for effective sim2real transfer, (iii) facilitates efficient parallelized offline simulation on accelerator hardware, (iv) supports real-time online processing capabilities that meet the latency and performance requirements of real-world systems. These claims are supported by experiments on two real-world systems. The pendulum swing-up task clearly demonstrates how neglecting delay simulation can impair

policy transfer, highlighting the need for delay-aware approaches, while the quadrotor task shows scalability to more complex robotic systems. The documentation, tutorials, and our open-source code can be found at `https://bheijden.github.io/rex/`. A video recording of the real-world experiments is available at `https://youtu.be/7j3OLUjTx_I`.

## 2 Related Work

**Sim2Real Frameworks** Sim2real frameworks such as Orbit (Mittal et al., 2023), Drake (Tedrake & the Drake Development Team, 2019), and EAGERx (van der Heijden et al., 2024b) facilitate the transfer of control policies from simulation to real-world settings. However, they generally do not include direct support for delay or dynamics identification from real-world data. Our framework addresses this gap by integrating these capabilities directly into the framework. Orbit utilizes Nvidia PhysX for parallelized simulations on accelerator hardware (NVIDIA, 2020). Our framework is based on JAX (Frostig et al., 2018) to support parallelized computation on accelerator hardware, while also enabling automatic differentiation. Moreover, our framework, like EAGERx (van der Heijden et al., 2024b), is not restricted to a specific simulator, as long as the simulator is compatible with JAX, such as Brax (Freeman et al., 2021) or the MJX extension of MuJoCo (Todorov et al., 2012). This flexibility enables users to select and extend engines as needed within the graph-based model. Tab. 1 provides a feature comparison between REX and related sim2real frameworks.

**Delay Estimation** System identification involves estimating the system's dynamics from input-output data and is a well-established area of research (Ljung, 1998). Traditional methods primarily focus on linear systems, often utilizing least-squares optimization techniques (Van Overschee & De Moor, 2012), while more recent efforts have extended to nonlinear systems (Nelles, 2020). Recent advances leverage the differentiability of general-purpose simulators to estimate complex system dynamics (Le Lidec et al., 2021; Heiden et al., 2022; Caluwaerts et al.,

|  | REX | Orbit | Drake | EAGERx |
|---|---|---|---|---|
| Multi-Sim Compatible | ✓ | ✗ | ✗ | ✓ |
| GPU Accelerated | ✓ | ✓ | ✗ | ✗ |
| Gradient Information | ✓ | ✗ | ✓ | ✗ |
| Delay Simulation | ✓ | ✓ | ✓ | ✓ |
| Delay Estimation | ✓ | ✗ | ✗ | ✗ |
| Dynamics Estimation | ✓ | ✗ | ✗ | ✗ |

Table 1: A feature comparison between REX and related sim2real frameworks.

2023). Our approach builds on these advancements by extending simulators with delay dynamics, allowing for the joint estimation of both system dynamics and delays. Instead of gradient-based methods, we use evolutionary strategies (Hansen, 2006), which we found to be less susceptible to local minima and better utilize the parallelism of modern hardware (Tang et al., 2022).

**Delay Simulation** Frameworks like Drake, EAGERx, and Orbit provide support for fixed delay simulation (Tedrake & the Drake Development Team, 2019; Mittal et al., 2023; van der Heijden et al., 2024b). Our framework, however, extends this capability by supporting stochastic delay simulation using Gaussian Mixture Models (GMMs). Additionally, it incorporates correlations between delays by considering the system's topology and communication structure during simulation. Although our framework allows for correlated delays, these delays are data-independent and do not change based on the simulated data. For example, even if an object detection algorithm takes longer to process when multiple objects are in view, our simulated delays remain the same regardless of the number of detected objects.

**Delay Compensation** Delay compensation in sim2real has been addressed through various methods. Algorithmic approaches for compensating delays have been proposed by Schuitema et al. (2010); Bouteiller et al. (2021). Other studies have enhanced sim2real performance by simulating delays during training and using a history of observations and actions as policy inputs (Tan et al., 2018; van der Heijden et al., 2024b). As part of their approach, Rudin et al. (2022) modified IsaacGym (Makoviychuk et al., 2021) to include a custom actuator model from (Hwangbo et al., 2019), which accounts for control signal delays caused by hardware/software layers. These methods teach policies to handle delays without compensating for them directly during real-world execution. Direct compensation techniques, such as the Smith predictor (Smith, 1957), have long been used in robotics to manage delays from sensors, actuators (Sherback et al., 2006; Augugliaro et al., 2014), and planning latency (Yang et al., 2020). In our work, we demonstrate that by

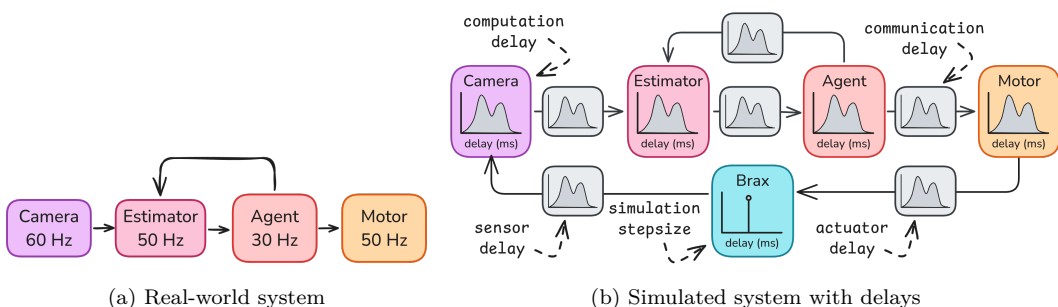

(a) Real-world system       (b) Simulated system with delays

Figure 2: Comparison between a real-world system setup and a simulated system with integrated delays. The real-world system (a) operates with different nodes at specified rates, while the simulated system (b) incorporates various types of delays to closely mimic real-world timing behaviors, including sensor, actuator, communication, and computation delays.

compensating for delays during execution, we can eliminate the need for delay simulation during training, resulting in a more efficient training process while maintaining high performance in real-world applications.

## 3 Our Sim2Real Framework

In this section, we present our framework for sim2real transfer in robotics, focusing on accurately modeling and compensating for the asynchronous interactions and delays encountered in real-world systems. In the following, we will first describe the graph-based architecture that facilitates asynchronous message passing and delay modeling. We will then detail the three runtime configurations designed for simulation, accelerated training, and real-time deployment. Finally, we will cover the integration of system identification techniques and delay compensation strategies to bridge the gap between simulation and reality.

### 3.1 Overview

The central element of our framework is the node, which represents a discrete unit of computation or sensing, operating asynchronously within the system. Nodes are designed to run at specified rates, processing inputs, maintaining state, and generating outputs. In our approach, both real-world and simulated systems are implemented as networks of these nodes, where communication occurs via directed edges, as shown in Fig. 2a. Each node's operation is defined by a step function that determines its behavior, transforming inputs into outputs. For example, nodes can represent various components such as cameras, agents, or motors, each handling specific tasks like sensing, control, or actuation. This modular design allows for flexible state, time, and action abstractions, supporting the modeling of complex interactions in a decentralized manner. Nodes are interconnected in a directed graph, facilitating asynchronous message passing and enabling nodes to operate at different rates. This design also enables the swapping of real-world nodes with simulated nodes, resulting in a unified software pipeline that can be used for sim2real transfer.

Asynchronous operations are inherent in real-world systems due to network transmission times, processing lags, or mechanical response times, which introduces delays into the system dynamics. To address this, we introduce a delay simulation model that captures both deterministic and stochastic delays, incorporating realistic timing behavior through delay distributions for communication, computation, sensor, and actuator delays. As shown in Fig. 2b, our model explicitly defines these delays as non-negative distributions, ensuring that the timing characteristics of the simulated environment closely match those of the real world. While this provides the structure for delay simulation, the challenge of estimating the correct delay parameters is addressed later in Sec. 3.3.

The framework supports multiple communication protocols to manage the flow of messages between nodes, allowing users to specify whether each communication channel should be blocking or non-blocking. A blocking channel ensures that a receiver node waits for the most recent message before processing, which minimizes latency in real-time systems by avoiding outdated information. However, blocking channels can introduce instability if delays cause unforeseen propagation through the graph; in such cases, non-blocking channels may

```
1  class Agent(BaseNode):
2    def init_params(self, rng, graph_state):
3      return PyTree(a=..., b=...)
4
5    def init_state(self, rng, graph_state):
6      return PyTree(x1=..., x2=...)
7
8    def init_output(self, rng, graph_state):
9      return PyTree(y1=..., y2=...)
10
11    # AOT jit-compile with graph.warmup()
12    def step(self, step_state):
13      ss = step_state  # Shorten name
14      # Read params, and current state
15      params, state = ss.params, ss.state
16      # Current episode, sequence, timestamp
17      eps, seq, ts = ss.eps, ss.seq, ss.ts
18      # Grab the data, and I/O timestamps
19      cam = ss.inputs['cam']  # connectd node
20      cam.data, cam.ts_send, cam.ts_recv
21      # Compute new state, and output
22      new_state = PyTree(x1=..., x2=...)
23      output = PyTree(y1=..., y2=...)
24      # Update step_state for next step call
25      new_ss = ss.replace(state=new_state)
26      return new_ss, output # Sends output
```

```
1  # Real-world nodes
2  cam = Camera(rate=60)
3  agent = Agent(rate=30)
4  motor = Motor(rate=50)
5  nodes = [cam, agent, motor]
6  # Connect
7  agent.connect(cam) # Async msging
8  motor.connect(agent, # Last 2 msgs
9               block=True, window=2)
10  # Runtime: WALL_CLOCK
11  # Used for real-time operation
12  graph = Graph(agent, nodes,
13               Clock.WALL_CLOCK,
14               RealTimeFactor.REAL_TIME)
15  # Ahead-of-time compilation of
16  # every node's .step() method
17  graph.warmup(devices=...)
18  # Run the graph at agent's rate
19  gs = graph.init() # Graph state
20  for i in range(100):
21    gs = graph.run(gs)
22  graph.stop()  # Halts all nodes
23  # Gather data (outputs, timings)
24  record = graph.get_record()
25  # Convert to data flow
26  df = record.to_graph()
```

```
1  # Simulation nodes & connections
2  cam = SimCam(rate=60, delay=Gauss(0.05, 0.01))
3  agent = Agent(rate=30, delay=Gauss(0.02, 0.01))
4  motor = SimMotor(rate=50, delay=Gauss(0.04, 0.01))
5  brax = Brax(rate=100, delay=Deterministic(0.01))
6  nodes = [brax, cam, agent, motor]
7  brax.connect(motor, delay=Gauss(0.01, 0.01))
8  cam.connect(brax, delay=Gauss(0.01, 0.01))
9  agent.connect(cam, delay=Gauss(0.01, 0.01))
10  motor.connect(agent, delay=Gauss(0.01, 0.01),
11               window=2, block=True)
12  # Runtime: SIMULATED (no throttling)
13  graph = Graph(agent, nodes, Clock.SIMULATED,
14               RealTimeFactor.FAST_AS_POSSIBLE)
15  graph.warmup(devices=...) # JIT compilation
16  gs = graph.init()  # Graph state
17  for i in range(100):  # Simulates 100 steps
18    gs = graph.run(gs)
19  graph.stop()  # Halts all nodes
20  # Simulated data flow to computation graph
21  cg = graph.get_record().to_graph().augment(nodes)
22  # Runtime: COMPILED (1000 parallel rollouts)
23  graph = CompiledGraph(agent, nodes, cg)
24  rngs = jax.split(jax.random.PRNGKey(0), num=1000)
25  gss = jax.vmap(graph.init)(rngs)  # 1000 states
26  rollout = jax.vmap(graph.rollout)(gss, rngs) # run
```

(a) Node definition      (b) Real-world runtime      (c) Simulation runtimes

Figure 3: Node definitions (a) use generic PyTrees that allow for compilation across different architectures for reduced latency. Examples of runtime configurations, showing different execution modes: WALL_CLOCK (b, Line 13) for real-time operation on physical hardware, SIMULATED (c, Line 13) for simulating without real-time constraints, and COMPILED (b, Line 23) for parallelized execution on accelerator hardware. Variable names and notations were slightly shortened for clarity and space.

be preferable. For example, an estimator node might opt for non-blocking behavior to continue predicting the system's state when sensor messages are delayed, allowing the controller to maintain responsive operation.

## 3.2 Runtimes

Our framework leverages JAX (Frostig et al., 2018) for efficient computation, utilizing its ability to perform just-in-time (JIT) compilation and automatic differentiation, which are crucial for high-performance machine learning applications. Nodes are defined using a generic interface, with parameters, states, and outputs specified using data structures that can be statically analyzed, as shown in Fig. 3a. This approach allows for ahead-of-time (AOT) compilation of the step method (Fig. 3a, Line 12) on various architectures, including CPUs and GPUs, thereby reducing latency. By compiling nodes in this manner, they can be seamlessly employed across different runtime modes without modification, ensuring flexibility and efficiency in both real-world and simulated environments. Our framework supports three distinct runtime modes, each tailored for different stages of development, training, and deployment: WALL_CLOCK, SIMULATED, and COMPILED.

The WALL_CLOCK runtime is designed for real-time execution on physical hardware, operating at real-time speed with each node's step function running asynchronously at its designated rate (Fig. 3b, lines 1-14). Nodes can be compiled to run on dedicated hardware resources such as separate CPU cores or accelerator hardware, minimizing latency (Fig. 3b, Line 17). After initializing the state of the graph, which aggregates the states of all nodes, the graph can be executed for a specified number of steps while recording the outputs and their corresponding timestamps (Fig. 3b, lines 19-24).

The SIMULATED runtime enables faster-than-real-time simulation, allowing for accelerated testing and development without real-time constraints. Message passing is based on simulated timestamps that are generated based on the communication protocol of every connection (blocking or non-blocking) and specified delay distributions, replicating real-world asynchronous effects (Fig. 3c, lines 1-14).

The COMPILED runtime further leverages accelerator hardware like GPUs or TPUs for parallelized execution by enabling the compilation of entire computation graphs into a single function. This makes this runtime suitable for tasks such as training RL policies and large-scale system identification that can leverage massive parallelism. Data flows from other runtimes (e.g., (Fig. 3b, Line 26)) are converted into a computation graph (Fig. 3c, Line 21) and compiled for parallel execution (Fig. 3c, lines 22-26), encoding the asynchronous effects

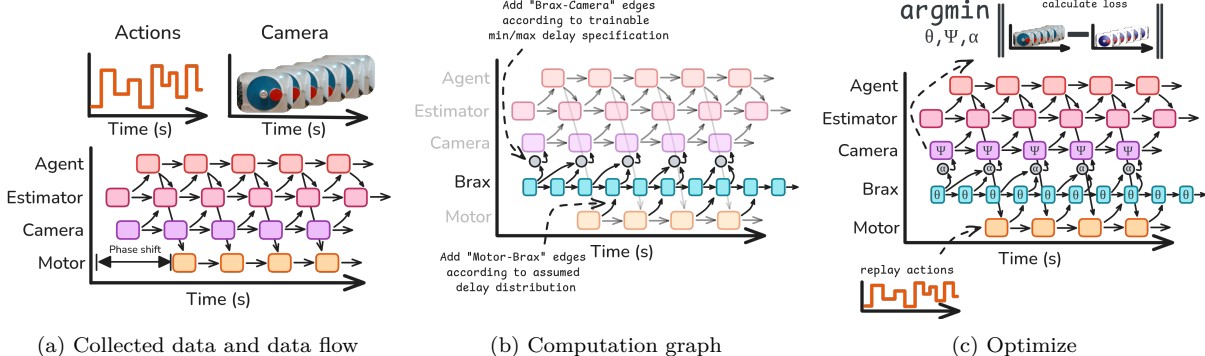

(a) Collected data and data flow          (b) Computation graph          (c) Optimize

Figure 4: System identification example applied to the system in Fig. 2a. (a) Data collection from the real-world system, including sensor data and timing information. (b) Construction of a computation graph that integrates the data flow with simulated nodes for dynamics and hidden delay identification. Motor-Brax edges are added based on a specified delay distribution, while Brax-camera edges follow a trainable min-max delay specification. (c) Optimization of simulation parameters and delays to minimize discrepancies between simulated and real-world behaviors, focusing on the delay interpolation parameter $\alpha$ and the parameters $\psi, \theta$ for the camera and Brax nodes.

of real-world interaction or simulated delays and enabling parallel execution on accelerator hardware. By supporting these three runtime modes, our framework provides comprehensive flexibility for a wide range of applications, from real-time deployment to parallelized system identification and policy training.

## 3.3   System Identification

System identification is crucial for minimizing the sim2real gap by ensuring that the simulated model closely mirrors the real-world system. Our framework facilitates this by identifying both the dynamics and delays inherent in real-world systems, allowing for more accurate simulation and effective delay compensation. In the following, we detail how to build and optimize a tailored computation graph from the real-world data collected to estimate system dynamics and delays (see Fig. 4). Data is collected from the real-world system using the `WALL_CLOCK` runtime, logging not only sensor and actuator data but also the timing information associated with message exchanges between nodes. This includes the timestamps for when a message is received, when a node begins processing, and when it sends out the output. Using this data, we construct a data flow graph that captures node interactions, including the precise timing of messages (see Fig. 4a, and Line 21 in Fig. 3c).

**Dynamics**   The data flow graph serves as a foundation for identifying the system's dynamics. One advantage of using a data flow graph is that it inherently represents asynchronous interactions and correctly encodes time-scale differences between nodes. Accounting for these asynchronous effects is essential, as they can significantly impact the identified system dynamics (Unbehauen & Rao, 1990). Given the data flow graph, our framework builds a tailored computation graph as follows. We augment the data flow graph with a simulator that models the system dynamics by adding simulator nodes at the desired simulation rate. Edges between simulation nodes and real-world-interacting nodes are introduced to pass the simulation state to the nodes that model real-world interactions (actuators, sensors, etc.), according to the assumed delay distributions, as shown in Fig. 4b. These delay distributions are either trainable or prespecified, as explained later in this section. By replaying actions through the computation graph and comparing the reconstructed outputs with the collected data, we optimize the simulator parameters to minimize a reconstruction loss. During this process, all parameters within the computation graph (e.g. simulator parameters or those in any other nodes), can be optimized. For instance, in the example shown in Fig. 4c, we simultaneously identify Brax's system parameters and the camera's parameters for angle-to-pixel conversion, but we could have also optimized for any other parameter in the graph, such as the motor's friction. The `COMPILED` runtime is particularly advantageous for this optimization process due to its ability to parallelize computations efficiently. We found evolutionary strategies effective for this task, as they leverage parallelism, constraint specification, and are less susceptible to local minima (Hansen, 2006; Tang et al., 2022; Lange, 2022a).

**Measurable Delays**   In addition to dynamics, our framework addresses delay estimation, distinguishing between directly measurable delays and hidden delays, such as those in actuators and sensors. Using the recorded timing data, we estimate the communication and computation delays of the system by fitting a Gaussian Mixture Model (GMM) to the measurable delay data using gradient descent. Details on the GMM fitting can be found in Appendix A. Typically, around a thousand samples are sufficient for fitting, which, depending on the system's rate, may require less than a minute of data collection. When sampling from the GMM, we clip the sampled values to be non-negative, as delays are inherently non-negative.

**Hidden Delays**   With hidden delays we mean delays that are not directly observable in the data flow graph, such as delays between the real world and sensors or actuators. While we support the addition of edges between simulator and real-world-interacting nodes based on prespecified delay distributions (e.g., motor-Brax connections in Fig. 4b), users can also introduce trainable delays to identify hidden delays (e.g., Brax-camera connections in Fig. 4b). Our approach requires specifying a minimum and maximum bound for each trainable delay, which we use to introduce additional edges that accommodate all possible communication patterns between two nodes under minimum and maximum delay conditions. We then introduce a trainable parameter $\alpha \in [0, 1]$ for each connection, allowing interpolation between the minimum and maximum scenarios. Different deterministic interpolation schemes, such as linear or zero-order hold, are currently supported to model various delay characteristics.

### 3.4   Delay Compensation

Once the system dynamics and delays are identified, the framework supports various strategies for delay compensation to enhance sim2real performance.

**Delay Simulation**   One straightforward strategy is to integrate the identified delay distributions into the simulation environment. This approach, referred to as delay simulation (Fig. 1c), allows the agent to learn policies that are delay aware. Notice that delays make the problem non-Markovian. To address this, a history of observations and actions can be stacked and used as input to the policy to restore the Markov property. This does make the learning problem more challenging, as the agent must learn to solve the task and handle delays simultaneously, as we will show in our experiments.

**Estimator**   While RL approaches often treat the environment as a black box, in sim2real scenarios, we can utilize the identified system dynamics and delays to design a model-based delay compensator that predicts the system's behavior during real-world execution. Inspired by a Smith Predictor (Smith, 1957) and shown in Fig. 1d, our strategy is to predict the state we expect when the corresponding command based on this state reaches the system. By knowing all delays, we can predict when a command will arrive and estimate the system state at that future time. Specifically, when a sensor captures an observation, we timestamp it and subtract the identified hidden delay $\tau_s$ to estimate the timestamp of the world's state the observation corresponds to, $t_s - \tau_s$. When the estimator processes the observation at $t_e$, it can determine when the resulting command will reach the system by adding the expected estimator-to-actuator latency, $\tau_a$, resulting in $t_e + \tau_a$. Thus, the estimator first updates the state up to $t_s - \tau_s$ and then predicts it forward to $t_e + \tau_a$ using the past control inputs and their estimated timestamps. We recommend using an Unscented Kalman Filter (UKF) (Julier & Uhlmann, 2004) for this task because it effectively handles non-differentiable and non-linear dynamics, while requiring only a small number of particles that can be efficiently evaluated in parallel (see Appendix B for more information). Additionally, in partially observable settings, a UKF can infer the hidden state of the system from observations and provide this state to the agent, enabling training in a fully observable, delay-free environment, which generally facilitates easier learning. In our experiments, we will evaluate the benefits of using such an estimator for delay compensation and compare the performance gains of delay compensation alone versus delay compensation with hidden state estimation.

### 3.5   Limitations

Our framework does not support running nodes on different machines; computations are restricted to different devices via JAX. This limits the ability to compile nodes for low-level controllers onboard a robot.

Additionally, JAX's Just-In-Time (JIT) compilation can lead to long compilation times, although recent updates with function caching have mitigated this to some extent.

The framework estimates hidden delays as deterministic, which is a reasonable assumption for many robotics applications. Nevertheless, stochastic delays can be modeled by adding variability to the deterministically identified delays, for example, to simulate jitter in sensor readings. Also, our approach requires setting minimum and maximum bounds for trainable delays, but we have found that using large bounds often yields good results. Furthermore, our delay simulation is state independent, meaning that while it accounts for the correlation and stochastic nature of delays, it does not adapt to the specific conditions or data of each simulation step. For instance, if an algorithm takes longer to process when there are multiple simulated objects in view, our approach would not capture this increase in processing time that would occur in a real-world scenario.

As systems scale to large configurations, efficiently parallelizing full asynchronicity for every node can become a challenge. To this end, we leverage the supergraph approach in (van der Heijden et al., 2024a) to efficiently parallelize graph-based simulations. Furthermore, the graph-based framework provides flexibility by allowing users to adjust the level of detail as needed for the task. For example, users may model entire robots as single nodes, focusing on interactions between them rather than internal asynchronicity, to maintain scalability in large-scale systems. The complexity of calculations within nodes is efficiently managed using JAX (Frostig et al., 2018), which enables scalable computations through its support for parallelization and distributed computing across multiple devices.

## 4 Experimental Evaluation

The main focus of this work is a sim2real framework that addresses asynchronous interactions in real-world systems by modeling delays and using real-world data for accurate system identification and reinforcement learning training. Our experiments are designed to validate the key claims made in Sec. 1 as follows. First, we identify the system dynamics and delays from real-world data, followed by a sim2real transfer evaluation using the identified system while using delay compensation techniques. We validate our approach on two distinct real-world systems: a pendulum swing-up and a quadrotor control task.

### 4.1 System Identification and Delay Estimation

To support the claim that our approach enables the identification of both dynamics and delays from real-world data, we present system identification and delay estimation results for the two selected systems.

**Pendulum** In contrast to the classic swing-up task (Brockman et al., 2016), which uses full state information, our setup relies solely on camera images of the pendulum. This task highlights the challenge of delay estimation and system identification from images. We apply an open-loop voltage sequence to the motor for 21 seconds while recording a stream of images from a RealSense d435i camera, in addition to the applied actions and corresponding timing information. Using this data, we construct a data flow graph that is augmented to form a computation graph, incorporating simulator nodes operating at 100 Hz. We introduce edges between the simulator nodes and the camera and motor via two trainable delays that assume a minimum and maximum delay of 0 to 50 ms, respectively. Images are first preprocessed through background subtraction and color thresholding to detect the center pixel coordinates of the red dot that marks the pendulum's mass.

The actions are then reapplied to the simulator, and we optimize the parameters to minimize the reconstruction error between predicted and actual pixel coordinates. Simultaneous optimization is performed on several parameters: the physics parameters of the Brax simulator (mass, length, friction, inertia, etc.), parameters for hidden camera and motor delays, and the parameters of an ellipse model (center, axes, rotation) that maps pixel coordinates to angles using the intuition that the pendulum's motion (as pixel coordinates) will be an ellipse when projected onto the camera plane. A UKF is employed for full state estimation and delay compensation, utilizing a lightweight dynamics model (see Appendix C). We use the Covariance Matrix Adaptation Evolution Strategy (CMA-ES) (Hansen, 2006) to optimize the 27 parameters by minimizing the

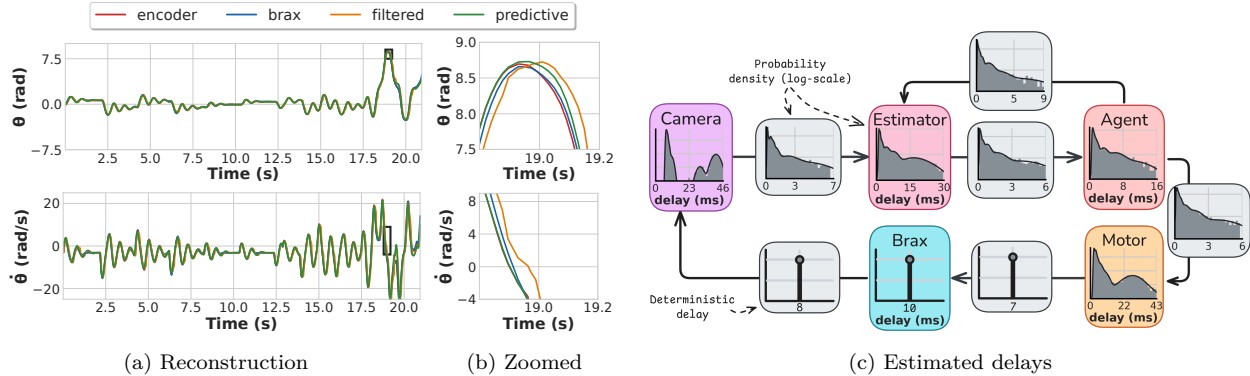

(a) Reconstruction        (b) Zoomed        (c) Estimated delays

Figure 5: Pendulum system identification and delay estimation. (a) Open-loop reconstruction of the angle ($\theta$) and angular velocity ($\dot{\theta}$) with the *brax* simulator, compared to ground-truth *encoder* data. (b) A zoomed view shows that the *predictive* UKF estimate that compensates for delays, outperforms the *filtered* estimate that does not. (c) Estimated GMM delay distributions and deterministic hidden delays for the camera and motor, with the grey area indicating the measured delay distribution and the black line showing the GMM fit.

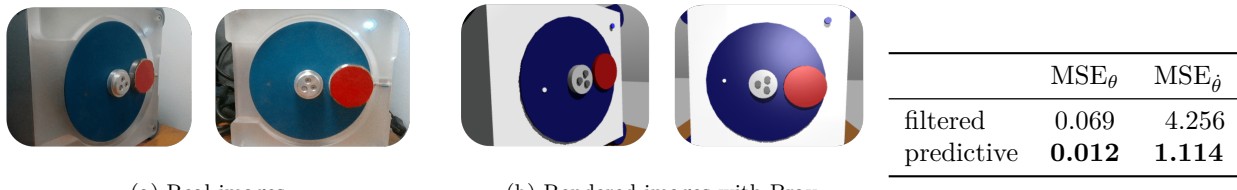

(a) Real images        (b) Rendered images with Brax

|  | MSE$_\theta$ | MSE$_{\dot{\theta}}$ |
|---|---|---|
| filtered | 0.069 | 4.256 |
| predictive | **0.012** | **1.114** |

Table 2: Mean squared error (MSE) with respect to ground-truth encoder data. Boldface indicates the best performance.

Figure 6: Comparison of real and rendered images of the pendulum from two different viewpoints. (a) shows actual images captured from side and frontal views. (b) shows the corresponding rendered images from the estimated poses.

reconstruction error between the predicted and measured pixel coordinates. See Appendix E for details on CMA-ES and the hyperparameter settings. Finally, we fit GMMs to estimate delay distributions for all measurable communication and computation delays.

The reconstructed angle and angular velocity from the simulator and estimator are shown in Fig. 5a, alongside the validation data obtained from the pendulum's encoder, which can be considered the ground truth with minimal delay. The open-loop reconstruction remains accurate over a 21-second time horizon. The identified delay distributions are illustrated in Fig. 5c, with a motor-to-Brax delay of approximately 7 ms and a Brax-to-camera delay of around 8 ms. The camera delay exhibits a multi-modal distribution, suggesting variability due to internal processing and shutter speed. The effectiveness of delay compensation is demonstrated in Fig. 5b by comparing the filtered and predictive estimates. The filtered estimate shows the UKF's state estimate plotted against the timestamp of when the action using the estimated state was applied to the simulator, resulting in a noticeable phase shift of around 50 ms. In contrast, the predictive estimate forecasts the filtered estimate forward, resulting in a lower mean squared error (MSE) for both the angle and angular velocity, as shown in Tab. 2. Finally, we use the identified system to render images from the estimated poses, as shown in Fig. 6. The comparison between real and rendered images from two different viewpoints qualitatively demonstrates the accuracy of the estimated system parameters.

**Quadrotor** Next, we identify the dynamics and delays of a quadrotor system using real-world data to demonstrate the applicability of our approach to higher-dimensional state-action spaces. The quadrotor's yaw is fixed, while the reference roll and pitch angles and the height setpoint are sent to a PID controller to maintain a circular flight path at a constant altitude. The PID controller converts the height setpoint to a thrust command, which, along with the roll and pitch commands, is sent to the Crazyflie. We record the actions, timing data, and state information captured by a motion capture (MoCap) system. Similar to the pendulum experiment, we construct a data flow graph that is augmented to form a computation

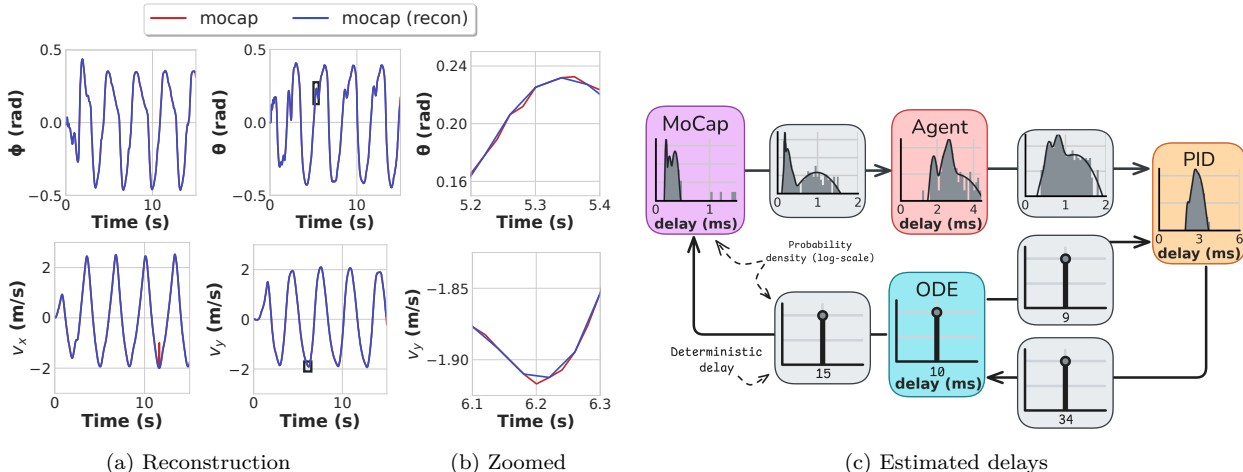

Figure 7: System identification and delay estimation for quadrotor control. (a) Open-loop reconstruction of roll, pitch, and velocities in body frame over 15 seconds (*recon*), showing the accuracy of the identified model compared to the MoCap data (*mocap*). (b) Zoomed view illustrates an accurate fit. (c) Estimated GMM delay distributions, with the grey area indicating the measured delay distribution and the black line showing the GMM fit.

graph with simulator nodes operating at 100 Hz. Edges are introduced between the simulator nodes and the MoCap and PID nodes, incorporating hidden delay nodes with a minimum and maximum delay of 0 to 50 ms, respectively. A dynamics model similar to that used in (Kooi & Babuska, 2021) is employed (see Appendix D). Simultaneous optimization is performed on the dynamics parameters (e.g., mass, drag, gain, time constants) and interpolation parameters for hidden delays between the dynamics model, MoCap, and PID controller, using CMA-ES (Hansen, 2006) to optimize the eight parameters by minimizing the reconstruction error between the predicted and measured quadrotor attitude and body frame velocities. See Appendix E for details on CMA-ES and the hyperparameter settings. We also fit GMMs to estimate delay distributions for all measurable communication and computation delays.

The results in Fig. 7a show accurate reconstruction of the quadrotor's states over 15 seconds. The identified delays are shown in Fig. 7c, with PID-to-ODE delay at 34 ms, ODE-to-PID delay at 9 ms, and ODE-to-MoCap delay at 15 ms. In the next section, we evaluate the advantage of delay-aware system identification for sim2real transfer by training policies with and without considering delays.

## 4.2 Sim2Real Transfer

To support the claim that our approach implements delay compensation techniques essential for effective sim2real transfer, we evaluate the sim2real performance of policies trained with and without delay compensation for the pendulum and quadrotor systems.

**Pendulum Swing-Up** This task highlights the challenge of delay compensation and partial observability in reinforcement learning. By demonstrating that neglecting delay simulation can impair policy transfer even in a seemingly simple scenario, we underscore the necessity of delay-aware approaches for more complex systems, where delays are inevitable and system dynamics are more intricate (Liu et al., 2019; Asaamoning et al., 2021; Lou et al., 2019; Peters et al., 2014). The pendulum task's simplicity effectively clarifies the importance of addressing delays in sim2real frameworks.

To investigate the impact of delays and partial observability on task complexity, we train pendulum swing-up policies using PPO (Schulman et al., 2017) under different conditions in simulation (see Appendix F for more details). We evaluate policies trained with full state information, stacked observations with and without delay simulation, and estimated full state information with simulated delays. As shown in Fig. 8a, policies with full state information achieve higher rewards and converge faster than those relying on stacked

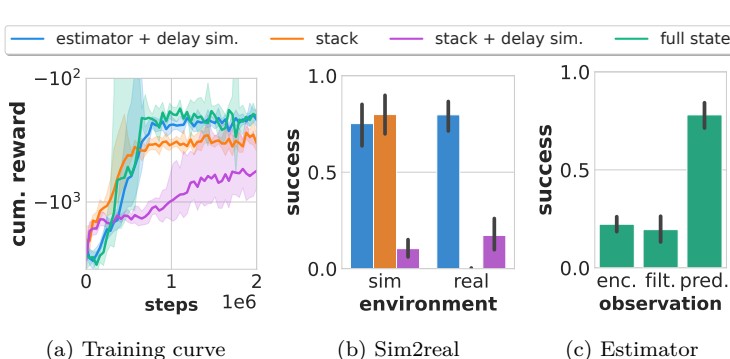

(a) Training curve      (b) Sim2real      (c) Estimator

Figure 8: Sim2real evaluation of policies trained under different delay and observation conditions for the pendulum swing-up task. (a) Training curves comparing policies with full state information and stacked observations. (b) Sim2real performance showing the percentage of time the pendulum remains upright (within $\pm 10°$ and $\pm 0.5\,\text{rad/s}$). (c) Performance of a policy using full state estimation with delay compensation, demonstrating the importance of delay compensation for steady performance.

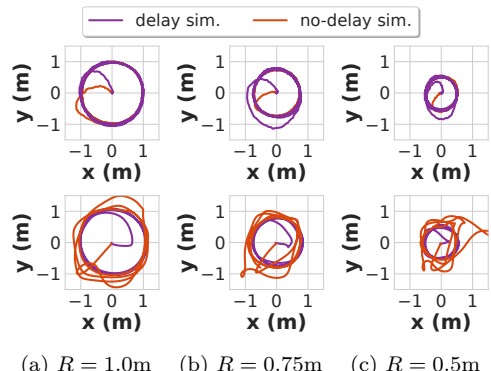

(a) $R = 1.0$m    (b) $R = 0.75$m    (c) $R = 0.5$m

Figure 9: Sim2real performance of quadrotor policies trained with and without delay simulation across different path radii. Top row: simulated path following at radii $R = 1.0$m, $R = 0.75$m, and $R = 0.5$m. Bottom row: real-world path following shows that delay simulation improves performance and stability, particularly at smaller radii where delays significantly impact control.

observations, especially when delays are present. This highlights the additional challenge introduced by delays and partial observability, beyond the complexity of the task itself.

Zero-shot evaluations on the real system show that policies trained solely with stacked observations fail to consistently swing up the pendulum, while the policy trained with delay simulation, delay compensation, and full state estimation achieves reliable swing-up, as demonstrated in Fig. 8b. Interestingly, even a policy trained on the full state without simulated delays can achieve consistent swing-up when real-world evaluation uses an estimator that compensates for delays and estimates the full state, as indicated by *pred.* in Fig. 8c. We assess the performance gains of delay compensation alone versus delay compensation with hidden state estimation, by evaluating the full state policy in two other scenarios: using the angle encoder, which provides full state information with negligible delay compared to camera images, and using the filtered state estimate from the UKF instead of the forward-predicted state. Both policies perform suboptimally, suggesting that both full state estimation and forward prediction are essential for reliable performance, as shown in Fig. 8c. A side-by-side comparison of the real-world swing-up performance of the different policies is available in the supplementary video.

**Path Following with a Quadrotor** We trained a quadrotor to fly a circular path at maximum speed with varying radii to assess the impact of delay simulation on sim2real performance. We used PPO (Schulman et al., 2017) to train policies using a reward function that penalizes path error and rewards high speeds along the path (see Appendix F for more details). In simulation, all policies achieved successful path following, with the no-delay policy reaching higher speeds and maintaining lower path errors due to its ability to fly more aggressively in the absence of simulated delays, as detailed in Tab. 3. However, in real-world tests, only the policy trained with delay simulation maintained stable flight; the no-delay policy exhibited oscillations around the target path. The performance gap widened at smaller radii, with the no-delay policy exhibiting highly unstable flight behavior at a radius of 0.5 m, demonstrating the critical role of delay-aware training for reliable real-world deployment, as shown in Fig. 9. A side-by-side comparison of the real-world flight performance of the two policies is available in the supplementary video.

## 4.3 Computational Runtime Analysis

To support our claim that the framework enables efficient parallelized simulation on accelerator hardware, we evaluated simulation speeds using the COMPILED runtime on an NVIDIA RTX 3070 Laptop GPU. The

| Radii (m) | Simulation | | | | Real-world | | | |
|---|---|---|---|---|---|---|---|---|
| | $v_{\text{path}}$ ($m/s$) | | $e_{\text{path}}$ ($m$) | | $v_{\text{path}}$ ($m/s$) | | $e_{\text{path}}$ ($m$) | |
| | delay | no-delay | delay | no-delay | delay | no-delay | delay | no-delay |
| 1.00 | 1.95 | **2.23** | 0.03 | **0.02** | 2.02 | **2.05** | **0.06** | 0.21 |
| 0.75 | 1.67 | **1.92** | 0.03 | **0.02** | **1.64** | 1.63 | **0.04** | 0.19 |
| 0.50 | 1.39 | **1.61** | **0.04** | **0.04** | **1.36** | 1.18 | **0.04** | 0.24 |

Table 3: Impact of delays on simulated vs. real-world performance across different path radii. $v_{\text{path}}$ denotes the average speed flown along the path, and $e_{\text{path}}$ represents the average error between the quadrotor's position and the target path. Boldface indicates the best performance in each category.

data flow was augmented with simulator nodes and subsequently parallelized to simulate delays according to real-world settings.

We measured the computation time for CMA-ES (Hansen, 2006) to converge during system identification for the pendulum and quadrotor tasks. For the pendulum, optimizing 27 parameters with a population size of 200 and a 21-second rollout per fitness evaluation (1,050 steps) led to convergence after 38 generations in 22.07 seconds, achieving 380k steps/s with a compilation time of 19.97 seconds. For the quadrotor, optimizing eight parameters under similar conditions but with a 15-second rollout (375 steps) resulted in convergence after 31 generations in 5.81 seconds, reaching 400k steps/s with a compilation time of 10.16 seconds. We also evaluated PPO training time using the implementation from (Lu et al., 2022): for the pendulum, training five policies in parallel with 64 environments reached 5 million steps in 77.1 seconds (325k steps/s), while for the quadrotor, training with 128 environments for 10 million steps completed in 29.8 seconds (336k steps/s), demonstrating the framework's efficiency in supporting rapid training on real-world tasks.

To isolate simulation speed from training overhead, we performed a parallelized rollout speed analysis (Fig. 10). The results show a linear relationship on a logarithmic scale, indicating that as the number of parallel environments doubles, the simulation speed also roughly doubles. An initial superlinear increase is observed, likely due to constant overheads being amortized over a larger number of parallel environments, resulting in more efficient resource utilization.

The simulation speed in our framework is determined by the computational workload of each node and the ability to parallelize their interactions. Our framework extends beyond standard simulations by modeling the asynchronous interaction between components, which are inherently challenging to parallelize efficiently (van der Heijden et al., 2024a). By demonstrating fast simulation speeds for the pendulum and quadrotor, we show that our framework achieves efficient runtime performance without introducing significant overhead beyond the computations within each node. If the simulation speed were slow, even with the simple dynamics of these systems, it would indicate a substantial fixed overhead from the framework.

To compare runtime performance with other sim2real frameworks, we evaluate the runtime of a common system across these frameworks. Specifically, we compare the pendulum system described in Appendix C with the pendulum examples in Drake (Tedrake & the Drake Development Team, 2019) and EAGERx (van der Heijden et al., 2024b), as all systems involve a simple pendulum with a two-dimensional state, ensuring a fair comparison. Since GPU parallelization is not supported in these frameworks, we measure the runtime performance of a single 20-second rollout on a CPU while applying random actions. Our results indicate that EAGERx achieves 0.28k steps/s, Drake achieves 60k steps/s, and our framework achieves 15k steps/s on a single CPU core. We attribute EAGERx's slower performance to the overhead of ROS communication between nodes, while Drake benefits from its optimized C++ backend. Our framework is significantly faster than EAGERx and, compared to Drake, can support GPU-based parallelized execution, which can further improve simulation speed, as demonstrated in Fig. 10.

To support the claim that our framework meets the latency and performance requirements for real-time online processing in real-world systems, we evaluate the latency of different components during real-world experiments. Our framework records timing information for each node, allowing us to estimate computation

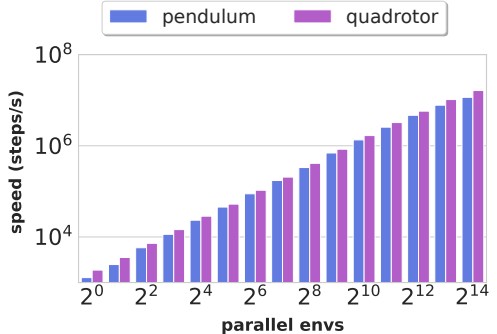

Figure 10: Simulation steps per second vs. number of parallel environments for a 200-step rollout on a GPU. The pendulum (20 ms/step) and quadrotor (25 ms/step) systems both demonstrate a linear scaling in simulation speed with increasing parallel environments.

| Node | Delay (ms) | Rate (Hz) | Device | Computation |
|---|---|---|---|---|
| | | **Pendulum** | | |
| camera | $9.8 \pm 8.0$ | 60 | $CPU_1$ | image-to-angle conv. |
| estimator | $3.5 \pm 5.2$ | 50 | $CPU_2$ | UKF update |
| agent | $1.7 \pm 2.3$ | 50 | GPU | Policy NN(64,64) |
| motor | $5.6 \pm 7.0$ | 50 | $CPU_3$ | Cmd to pendulum |
| | | **Quadrotor** | | |
| mocap | $0.3 \pm 0.1$ | 50 | $CPU_1$ | Read quadrotor pose |
| agent | $2.6 \pm 0.5$ | 25 | GPU | Policy NN(64,64) |
| pid | $2.8 \pm 0.3$ | 50 | $CPU_2$ | Cmds to quadcopter |

Table 4: Delay statistics for Quadrotor and Pendulum nodes, including delay (mean ± std.) in ms, rate in Hz, device type, and a description of each computation. Subscripts indicate dedicated CPU cores. NN denotes a neural network with layer sizes in parentheses.

and communication delay statistics across both the pendulum and quadrotor systems, as visualized in Fig. 5c and Fig. 7c. The mean and standard deviation of delays for each node's periodic computation are calculated, providing insights into system performance. By dedicating specific CPU cores to each node, we bypass the Python Global Interpreter Lock (GIL), enabling concurrent execution. Additionally, we use the GPU to accelerate policy inference in the agent node. This approach results in low latency across the system. Unexpectedly, the motor node in the pendulum system exhibited large delays, likely due to the hardware's slow response time while servicing ROS (Quigley et al., 2009) service calls. As expected, the camera node had the longest delays, attributed to the time required for image retrieval and processing to convert images to angles.

In summary, our evaluation demonstrates that our approach effectively identifies both dynamics and delays from real-world data, compensates for delays to improve sim2real transfer, and facilitates efficient parallelized simulation on accelerator hardware. At the same time, our approach meets the latency and performance requirements for real-time online processing, supporting all four key claims.

## 5 Conclusion

In this paper, we presented a novel framework, REX (Robotic Environments with jaX), for sim2real transfer that introduces a graph-based simulation model incorporating latency effects, optimized for parallelization on accelerator hardware. Our approach models asynchronous, hierarchical systems by explicitly representing computation, communication, actuation, and sensing delays. This enables the simultaneous estimation of system dynamics and delays using real-world data, effectively minimizing the sim2real gap. We implemented and evaluated our approach on two real-world robotic systems, demonstrating its ability to support rapid training while maintaining high fidelity to real-world conditions. The experiments suggest that our framework not only improves the accuracy of policy transfer by reducing the impact of delays and partial observability but also enhances simulation efficiency by leveraging hardware acceleration.

For future work, we aim to extend the framework to support estimating stochastic hidden delays, which could further reduce the sim2real gap by more accurately capturing real-world uncertainties. Additionally, we plan to enhance the framework's scalability and real-world applicability by enabling distributed computing across multiple machines, beyond the current capability of utilizing different devices via JAX.

## 6 Acknowledgments

This work was funded in part by the EU's H2020 OpenDR project (grant No 871449).

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

## A  Measurable Delay Fitting with Gaussian Mixture Models

Let $\{x_i\}_{i=1}^N$ represent the observed delays as one-dimensional data points. We consider a Gaussian Mixture Model (GMM) with $K$ components, where each component $k$ is characterized by three parameters: $(\pi_k, \mu_k, \sigma_k)$. Specifically,

- $\pi_k$: the mixing weight of component $k$, satisfying $\sum_{k=1}^K \pi_k = 1$ and $\pi_k \geq 0$,

- $\mu_k$: the mean of the Gaussian component,

- $\sigma_k$: the standard deviation of the Gaussian component.

The full set of parameters of the GMM is denoted as $\boldsymbol{\theta} = \{\pi_k, \mu_k, \sigma_k\}_{k=1}^K$, which includes all mixing weights, means, and standard deviations. The probability density function for a single data point $x_i$ under this model is expressed as:

$$p(x_i \mid \boldsymbol{\theta}) = \sum_{k=1}^K \pi_k \mathcal{N}(x_i \mid \mu_k, \sigma_k^2),$$

where $\mathcal{N}(x_i \mid \mu_k, \sigma_k^2)$ is the probability density function of a normal distribution with mean $\mu_k$ and variance $\sigma_k^2$. To estimate the parameters of the GMM, we minimize the negative log-likelihood of the observed delays:

$$\mathcal{L}(\boldsymbol{\theta}) = -\sum_{i=1}^N \log \left( \sum_{k=1}^K \pi_k \mathcal{N}(x_i \mid \mu_k, \sigma_k^2) \right).$$

The optimization is performed using a gradient-based approach. Specifically, the parameters $\boldsymbol{\theta}$ are updated iteratively using a standard solver such as the Adam optimizer (Kingma, 2014), which adjusts the parameters to minimize $\mathcal{L}(\boldsymbol{\theta})$. To improve numerical stability during optimization, we normalize the data before fitting the GMM. The observed delays $\{x_i\}$ are transformed into normalized delays $\{x_i'\}$ as:

$$x_i' = \frac{x_i - \mu_x}{\max(\sigma_x, \epsilon)},$$

where $\mu_x$ and $\sigma_x$ are the mean and standard deviation of the observed delays, and $\epsilon$ is a small constant (e.g., $10^{-7}$) to avoid division by zero. The GMM is then fit to the normalized dataset $\{x_i'\}_{i=1}^N$, and the negative log-likelihood is computed accordingly. After fitting the model, the parameters are denormalized to map back to the original data range with:

$$\tilde{\mu}_k = \mu_k \cdot \sigma_x + \mu_x,$$
$$\tilde{\sigma}_k = \sigma_k \cdot \sigma_x.$$

This approach ensures numerical stability during optimization while providing parameters $\tilde{\mu}_k$ and $\tilde{\sigma}_k$ in the original scale of the observed delays.

## B  Unscented Kalman Filter

The Unscented Kalman Filter (UKF) is a state estimation algorithm designed for nonlinear systems. It employs a deterministic sampling technique to approximate the mean and covariance of the state distribution. This approach is more accurate than linearization-based methods, such as the Extended Kalman Filter (EKF), for highly nonlinear systems (Simon, 2006). Since the UKF relies on sampling rather than linearization, it can effectively handle non-differentiable functions. The number of sigma points, $2n + 1$, is determined by the state dimensionality $n$ and is typically much smaller than the number of particles required for a particle filter. Additionally, the evaluation of sigma points can be efficiently parallelized. The sigma points are calculated to capture the mean and covariance of the state distribution and we follow the procedure in (Van Der Merwe, 2004) to generate $2n + 1$ sigma points. Given the current state mean $\mathbf{x}$ and covariance $\mathbf{P}$, the scaling parameter $\lambda = \alpha^2(n + \kappa) - n$ is computed, where $\alpha$ controls the spread of the sigma points, $\kappa$ adjusts scaling, and $\beta$ incorporates prior knowledge (e.g., $\beta = 2$ for Gaussian distributions). The sigma points are then determined as:

$$\chi_0 = \mathbf{x}, \quad \chi_i = \mathbf{x} \pm \sqrt{n + \lambda} \cdot [\mathbf{P}]_i, \quad i = 1, \ldots, n$$

where $\sqrt{n + \lambda} \cdot [\mathbf{P}]_i$ is the $i$-th column of the Cholesky decomposition of $(n+\lambda)\mathbf{P}$. The corresponding weights for mean and covariance are:

$$w_0^{(m)} = \frac{\lambda}{n + \lambda}, \quad w_0^{(c)} = w_0^{(m)} + (1 - \alpha^2 + \beta), \quad w_i^{(m)} = w_i^{(c)} = \frac{1}{2(n + \lambda)}, \quad i = 1, \ldots, 2n$$

The filter is initialized with an initial state mean $\mathbf{x}_0$ and covariance $\mathbf{P}_0$. The state transition function $f(\cdot)$ and observation function $h(\cdot)$ are nonlinear, describing the process dynamics and measurements, respectively. Process and measurement noise are assumed to be additive, Gaussian, and uncorrelated, with zero mean and covariance matrices $\mathbf{Q}$ and $\mathbf{R}$, respectively. Actions commanded at time $t-1$ will affect the state at time $t + \tau_a$, where $\tau_a$ is the actuation delay. Similarly, the most recent measurement available at time $t$ is from time $t - \tau_s$, where $\tau_s$ is the sensor delay. Hence, the indexing of the measurements and actions is shifted by the sensor and actuation delays. For simplicity, we assume that the sensor delay $\tau_s$ and actuation delay $\tau_a$ are constant and a multiple of the timestep. Furthermore, we consider time-invariant noise distributions, process models, and measurement models. Note, however, that the UKF can be extended to handle time-varying noise covariances, models, variable delays, and non-uniform timesteps, but we do not consider them here for notational clarity.

The UKF estimation step at time $t$ is outlined in Alg. 1. In the prediction and update steps, the UKF updates its prior state estimate from $t - 1 - \tau_s$ to the last measurement time, $t - \tau_s$, using the most recent measurement available at $t - \tau_s$. Next, the UKF forward-predicts the state to $t + \tau_a$, incorporating the commanded actions available up to $t - 1 + \tau_a$ to account for the combined sensor and actuation delays.

---

**Algorithm 1:** UKF Estimation Step with Delay Compensation

---

**Input:** Process noise $\mathbf{Q}$, measurement noise $\mathbf{R}$, weights $w_i^{(m)}, w_i^{(c)}$, process model $f(\cdot)$, measurement model $h(\cdot)$, sensor delay $\tau_s$, actuation delay $\tau_a$

**Input:** Previous state mean $\mathbf{x}_{t-1-\tau_s}$, covariance $\mathbf{P}_{t-1-\tau_s}$, new measurement $\mathbf{z}_{t-\tau_s}$, action sequence $\{\mathbf{u}_k\}_{k=t-1-\tau_s}^{t-1+\tau_a}$

**Output:** Estimated state mean $\mathbf{x}_{t-\tau_s}$, covariance $\mathbf{P}_{t-\tau_s}$, forward-predicted state mean $\mathbf{x}_{t+\tau_a}$, covariance $\mathbf{P}_{t+\tau_a}$

1   **Calculate Sigma Points:**

2   Compute sigma points $\{\chi_i^{(t-1-\tau_s)}\}$ from current state mean $\mathbf{x}_{t-1-\tau_s}$ and covariance $\mathbf{P}_{t-1-\tau_s}$.

3   **Prediction Step:**

4   Propagate sigma points through process model with actions: $\chi_i^- = f(\chi_i^{(t-1-\tau_s)}, \mathbf{u}_{t-1-\tau_s})$.

5   Compute predicted mean: $\mathbf{x}_{t-\tau_s}^- = \sum_i w_i^{(m)} \chi_i^-$.

6   Compute predicted covariance: $\mathbf{P}_{t-\tau_s}^- = \sum_i w_i^{(c)} (\chi_i^- - \mathbf{x}_{t-\tau_s}^-)(\chi_i^- - \mathbf{x}_{t-\tau_s}^-)^T + \mathbf{Q}$.

7   **Update Step:**

8   Propagate sigma points through measurement model: $\mathbf{z}_i = h(\chi_i^-)$.

9   Compute predicted measurement mean: $\mathbf{z}_{t-\tau_s}^- = \sum_i w_i^{(m)} \mathbf{z}_i$.

10   Compute innovation covariance: $\mathbf{S} = \sum_i w_i^{(c)} (\mathbf{z}_i - \mathbf{z}_{t-\tau_s}^-)(\mathbf{z}_i - \mathbf{z}_{t-\tau_s}^-)^T + \mathbf{R}$.

11   Compute cross-covariance: $\mathbf{C} = \sum_i w_i^{(c)} (\chi_i^- - \mathbf{x}_{t-\tau_s}^-)(\mathbf{z}_i - \mathbf{z}_{t-\tau_s}^-)^T$.

12   Compute Kalman gain: $\mathbf{K} = \mathbf{C}\mathbf{S}^{-1}$.

13   Update state mean: $\mathbf{x}_{t-\tau_s} = \mathbf{x}_{t-\tau_s}^- + \mathbf{K}(\mathbf{z}_{t-\tau_s} - \mathbf{z}_{t-\tau_s}^-)$.

14   Update covariance: $\mathbf{P}_{t-\tau_s} = \mathbf{P}_{t-\tau_s}^- - \mathbf{K}\mathbf{S}\mathbf{K}^T$.

15   **Forward-prediction Step:**

16   **for** $k = t - \tau_s, \ldots, t - 1 + \tau_a$ **do**

17      Compute sigma points $\{\chi_i^{(k)}\}$ from state mean $\mathbf{x}_k$ and covariance $\mathbf{P}_k$.

18      Propagate sigma points through process model with actions: $\chi_i^- = f(\chi_i^{(k)}, \mathbf{u}_k)$.

19      Compute forward-predicted mean: $\mathbf{x}_{k+1} = \sum_i w_i^{(m)} \chi_i^-$.

20      Compute forward-predicted covariance: $\mathbf{P}_{k+1} = \sum_i w_i^{(c)} (\chi_i^- - \mathbf{x}_{k+1})(\chi_i^- - \mathbf{x}_{k+1})^T + \mathbf{Q}$.

21   **end**

22   **return** $\mathbf{x}_{t-\tau_s}, \mathbf{P}_{t-\tau_s}, \mathbf{x}_{t+\tau_a}, \mathbf{P}_{t+\tau_a}$

---

## C   Pendulum Dynamics

The pendulum system is modeled by a second-order ordinary differential equation (ODE). The state $\mathbf{x} = (\theta, \dot{\theta})$ represents the angle $\theta$ and angular velocity $\dot{\theta}$. The control input $u$ represents the applied voltage. The angular acceleration $\ddot{\theta}$ is given by:

$$\ddot{\theta} = \frac{u\frac{K}{R} + mgl\sin(\theta) - b\dot{\theta} - \dot{\theta}\frac{K^2}{R} - c\,\mathrm{sign}(\dot{\theta})}{J},$$

where $J$ is the moment of inertia, $m$ the mass, $l$ the pendulum length, $b$ the damping coefficient, $K$ the motor constant, $R$ the motor resistance, $c$ the static friction, and $g = 9.81\,\mathrm{m/s}^2$ the gravitational acceleration.

Dynamics are simulated using a fourth-order Runge-Kutta integration method with a fixed time step of 0.01 seconds. All parameters ($J$, $m$, $l$, $b$, $K$, $R$, $c$) are identified experimentally in this paper except for the gravitational acceleration $g$, together with any additional parameters required for hidden delay estimation.

## D   Quadrotor Dynamics

The dynamics model, similar to that in Kooi & Babuska (2021), is used to simulate the quadrotor's motion. The dynamics are divided into three components: rotational, translational, and motor dynamics. The state is represented by the position $p = (x, y, z)$, velocity $v = (\dot{x}, \dot{y}, \dot{z})$, attitude $\eta = (\phi, \theta, \psi)$, and thrust state $\Omega$. The control inputs are the reference pulse-width modulation (PWM) motor signal $\Theta$ and two reference angles $\phi_{\text{ref}}$ and $\theta_{\text{ref}}$. Yaw dynamics are neglected, with yaw angle assumed constant at $\psi = 0$. The rotational dynamics are approximated by a first-order system for the attitude angles $\phi$ and $\theta$:

$$\dot{\phi} = \frac{k_c(\phi_{\text{ref}} - \phi)}{\tau_c} \qquad\qquad \dot{\theta} = \frac{k_c(\theta_{\text{ref}} - \theta)}{\tau_c}$$

where the same $k_c$ and $\tau_c$ are used for the rotation in both angle directions due to the system's symmetry. These dynamics comprise the quadrotor's closed-loop onboard control of the attitude angles. The total thrust generated by the quadrotor's motors is modeled as a first-order system:

$$\dot{\Omega} = a_{\text{m}}\Omega + b_{\text{m}}\Theta \qquad\qquad f_{\text{thrust}} = c_{\text{m}}\Omega + d_{\text{m}}\Theta$$

where $a_{\text{m}}$, $b_{\text{m}}$, $c_{\text{m}}$, and $d_{\text{m}}$ are motor-specific constants. The drag force acting on the quadrotor is given by:

$$f_{\text{drag}} = - \begin{bmatrix} \kappa_{xy}\omega & 0 & 0 \\ 0 & \kappa_{xy}\omega & 0 \\ 0 & 0 & \kappa_z\omega \end{bmatrix} v_b,$$

where $\kappa_{xy}$ and $\kappa_z$ are drag coefficients, $\omega$ is the rotor speed, and $v_b$ is the body-frame velocity. The body-frame velocity is calculated as $v_b = R^\top v^\top$, where $R$ is the rotation matrix from the body frame to the world frame. The rotor speed is approximated using the following relationships (see Förster (2015) for more details) between the effective PWM signal $\Theta_{\text{eff}}$, rotor speed $\omega$, and total thrust $f_{\text{thrust}}$:

$$f_{\text{thrust}} = 4\big(a_{\text{p}}\Theta_{\text{eff}}^2 + b_{\text{p}}\Theta_{\text{eff}} + c_{\text{p}}\big) \qquad\qquad \omega = 4\big(a_{\text{r}}\Theta_{\text{eff}} + b_{\text{r}}\big)$$

where $a_{\text{p}}$, $b_{\text{p}}$, and $c_{\text{p}}$ are PWM constants, $a_{\text{r}}$ and $b_{\text{r}}$ are rotor constants, and the factor 4 accounts for the quadrotor's four rotors. The translational dynamics are:

$$\dot{v} = \frac{1}{m}R\big([0, 0, f_{\text{thrust}}]^\top + f_{\text{drag}}\big) - [0, 0, g]^\top,$$

where $m$ is the quadrotor's mass, and $g$ is the gravitational constant. Dynamics are simulated using a fourth-order Runge-Kutta integration method with a fixed time step of 0.01 seconds.

The motor, PWM, and rotor constants are specific to the motor and require additional sensors for accurate identification. We use the experimentally identified values from Förster (2015) for the Crazyflie 2.0 quadrotor, as done in Kooi & Babuska (2021):

$$a_{\text{m}} = -15.47, \qquad b_{\text{m}} = 1.0, \qquad c_{\text{m}} = 1.43 \times 10^{-4}, \qquad d_{\text{m}} = 2.89 \times 10^{-7},$$
$$a_{\text{p}} = 2.13 \times 10^{-11}, \qquad b_{\text{p}} = 1.03 \times 10^{-6}, \qquad c_{\text{p}} = 5.49 \times 10^{-4},$$
$$a_{\text{r}} = 0.041, \qquad b_{\text{r}} = 380.83.$$

The remaining parameters ($m$, $k_c$, $\tau_c$, $\kappa_{xy}$, $\kappa_z$) are experimentally identified in this paper, along with any additional parameters required for hidden delay estimation.

# E    Covariance Matrix Adaptation Evolution Strategy

The Covariance matrix adaptation evolution strategy (CMA-ES) algorithm is a stochastic, derivative-free optimization method well-suited for non-linear or non-convex problems (Hansen, 2006). It evolves a population of solutions by sampling from a multivariate normal distribution, adapting the covariance matrix and step size to guide search directions efficiently.

The algorithm steps are outlined in Alg. 2, where the key hyperparameters are defined as follows. First, the total number of iterations or generations, denoted as $G$, determines how long the algorithm runs. Each iteration involves evaluating a population of candidate solutions, the size of which is specified by $\lambda$. From this population, the top-performing $\mu$ solutions are selected for recombination, with the condition $\mu \le \lambda$. The learning process also depends on several adaptation rates: $c_\sigma$ and $d_\sigma$ control the adaptation of the step size, while $c_c$, $c_1$, and $c_\mu$ influence the covariance matrix updates, ensuring efficient exploration of the search space. The initial mean vector, $\mathbf{m}^{(0)} \in \mathbb{R}^n$, represents the initial estimate in the search space, while the initial step size, $\sigma^{(0)}$, scales the search distribution. To ensure isotropic sampling at the outset, the initial covariance matrix, $\mathbf{C}^{(0)}$, is typically set to the identity matrix, $\mathbf{I}$. Note that it is common practice to search over a normalized space, where the search distribution is isotropic and centered at the origin, to improve numerical stability. Hence, the initial mean vector and covariance matrix are initialized to zero and the identity matrix, respectively.

---

**Algorithm 2:** CMA-ES (Covariance Matrix Adaptation Evolution Strategy)

---

**Input:** Population size $\lambda$, initial mean $\mathbf{m}^{(0)}$, initial step size $\sigma^{(0)}$, initial covariance matrix $\mathbf{C}^{(0)} = \mathbf{I}$, weights $w_1, \ldots, w_\mu$
**Output:** Optimized solution $\mathbf{m}^*$

1   **Initialize:**

2   Set $w_i \leftarrow \dfrac{\log(\mu + \frac{1}{2}) - \log(i)}{\sum_{j=1}^{\mu} \left( \log(\mu + \frac{1}{2}) - \log(j) \right)}$ for $i = 1, \ldots, \mu$

3   Normalize weights: $w_i \leftarrow \dfrac{w_i}{\sum_{j=1}^{\mu} w_j}$

4   Set $\mathbf{p}_\sigma^{(0)} \leftarrow \mathbf{0}$, $\mathbf{p}_c^{(0)} \leftarrow \mathbf{0}$, $\mu_{\text{eff}} \leftarrow (\sum_{i=1}^{\mu} w_i)^2 / \sum_{i=1}^{\mu} w_i^2$

5   Set learning rates $c_\sigma, c_c, c_1, c_\mu$, damping factor $d_\sigma$, and chi constant $\chi_n \leftarrow \sqrt{n}(1 - \frac{1}{4n} + \frac{1}{21n^2})$

6   **Generation loop:**

7   **for** $g = 0, 1, \ldots,$ *G-1* **do**

8        **for** $k \leftarrow 1$ *to* $\lambda$ **do**

9            Sample: $\mathbf{x}_k \sim \mathcal{N}(\mathbf{m}, \sigma^2 \mathbf{C})$

10           Evaluate fitness: $f(\mathbf{x}_k)$

11       **end**

12       Sort $\mathbf{x}_1, \ldots, \mathbf{x}_\lambda$ by fitness, and select the $\mu$ best solutions

13       Compute new mean: $\mathbf{m}^{(g+1)} \leftarrow \sum_{i=1}^{\mu} w_i \mathbf{x}_i$

14       **Update step-size evolution path:**

15       $\mathbf{p}_\sigma^{(g+1)} \leftarrow (1 - c_\sigma)\mathbf{p}_\sigma^{(g)} + \sqrt{c_\sigma(2 - c_\sigma)\mu_{\text{eff}}} \cdot \mathbf{C}^{-\frac{1}{2}} \frac{\mathbf{m}^{(g+1)} - \mathbf{m}^{(g)}}{\sigma^{(g)}}$

16       $h_\sigma \leftarrow \|\mathbf{p}_\sigma^{(g+1)}\| / \sqrt{1 - (1 - c_\sigma)^{2(g+1)}} < (1.4 + \frac{2}{n+1}) \cdot \chi_n$

17       **Update covariance evolution path:**

18       $\mathbf{p}_c^{(g+1)} \leftarrow (1 - c_c)\mathbf{p}_c^{(g)} + h_\sigma \cdot \sqrt{c_c(2 - c_c)\mu_{\text{eff}}} \cdot \frac{\mathbf{m}^{(g+1)} - \mathbf{m}^{(g)}}{\sigma^{(g)}}$

19       **Update covariance matrix:**

20       $\mathbf{C}^{(g+1)} \leftarrow (1 - c_1 - c_\mu)\mathbf{C}^{(g)} + c_1 \mathbf{p}_c^{(g+1)} \mathbf{p}_c^{(g+1)^\top} + c_\mu \sum_{i=1}^{\mu} w_i \mathbf{y}_i \mathbf{y}_i^\top$

21       where $\mathbf{y}_i = \frac{\mathbf{x}_i - \mathbf{m}^{(g)}}{\sigma^{(g)}}$

22       **Update step size:**

23       $\sigma^{(g+1)} \leftarrow \sigma^{(g)} \cdot \exp\left( \frac{c_\sigma}{d_\sigma} \cdot \left( \frac{\|\mathbf{p}_\sigma^{(g+1)}\|}{\chi_n} - 1 \right) \right)$

24  **end**

25  **return** $\mathbf{m}^* \leftarrow \mathbf{m}^{(g+1)}$

---

The table below summarized the hyperparameters used in the CMA-ES algorithm for the Pendulum and Quadrotor tasks in this paper.

| Hyperparameter | Name | Pendulum | Quadrotor |
|---|---|---|---|
| $G$ | Generations | 40 | 100 |
| $\lambda$ | Population size | 200 | 200 |
| $\mu$ | Selected solutions | 20 | 20 |
| $c_\sigma$ | Step-size learning rate | 0.40 | 0.65 |
| $c_c$ | Covariance learning rate | 0.14 | 0.68 |
| $c_1$ | Rank-1 update rate | 0.0024 | 0.19 |
| $c_\mu$ | Rank-$\mu$ update rate | 0.038 | 0.27 |
| $d_\sigma$ | Step-size damping | 1.40 | 2.41 |
| $\mathbf{m}^{(0)}$ | Initial mean vector | $\mathbf{0}$ | $\mathbf{0}$ |
| $\sigma^{(0)}$ | Initial step size | 0.4 | 0.4 |
| $\mathbf{C}^{(0)}$ | Initial covariance matrix | $\mathbf{I}$ | $\mathbf{I}$ |

## F   Proximal Policy Optimization

Proximal Policy Optimization (PPO) is a reinforcement learning algorithm designed to optimize policies by maximizing the expected return (Schulman et al., 2017). However, reproducing PPO's results can be challenging due to its sensitivity to hyperparameters and implementation details (Huang et al., 2023). For this reason, we provide the exact PPO implementation used in this paper in the supplementary material. Our implementation is largely based on (Lu et al., 2022), which itself builds upon Huang et al. (2022).

The table below summarizes the hyperparameter settings used for the two tasks in this paper:

| Hyperparameter | Name | Pendulum | Quadrotor |
|---|---|---|---|
| $T$ | Total timesteps | $2 \times 10^6$ | $10 \times 10^6$ |
| $\eta$ | Learning rate | $3.26 \times 10^{-4}$ | $9.23 \times 10^{-4}$ |
| $n_{\text{envs}}$ | Number of environments | 128 | 128 |
| $n_{\text{steps}}$ | Number of steps per update | 32 | 64 |
| $E$ | Number of epochs | 8 | 16 |
| $n_{\text{minibatch}}$ | Number of minibatches | 16 | 8 |
| $\gamma$ | Discount factor | 0.9939 | 0.9844 |
| $\lambda_{\text{GAE}}$ | GAE lambda | 0.971 | 0.939 |
| $\epsilon_{\text{clip}}$ | Clipping epsilon | 0.164 | 0.131 |
| $\alpha_{\text{ent}}$ | Entropy coefficient | 0.01 | 0.01 |
| $\alpha_{\text{vf}}$ | Value function coefficient | 0.802 | 0.756 |
| $g_{\text{max}}$ | Max gradient norm | 0.963 | 0.76 |
| $n_{\text{hidden}}$ | Number of hidden layers | 2 | 2 |
| $h_{\text{units}}$ | Number of hidden units | 64 | 64 |
| $\phi_{\text{hidden}}$ | Hidden activation | tanh | tanh |
| $\sigma_{\text{ind}}$ | State-independent action noise | True | True |
| Squash | Action squashing | True | True |
| $\eta_{\text{anneal}}$ | Anneal learning rate | False | False |
| Norm | Normalize environment | True | True |

