# OpenReview forum: "REX: GPU-Accelerated Sim2Real Framework with Delay and Dynamics Estimation"
_TMLR — Accepted by TMLR_

### Review · Reviewer_qxE9 · 2024-11-17

**Summary Of Contributions:**

This paper addresses the challenges of sim2real transfer caused by unmodeled dynamics and latencies in real-world robotic systems, proposing a novel framework called REX to incorporate latency effects and optimize for parallelization accelerator hardware. The framework is validated on two systems: the real-world pendulum swing-up task and a quadrocopter path following task. The authors demonstrate the effectiveness of their REX framework in improving sim2real performance.

**Audience:**

Yes

**Broader Impact Concerns:**

No concerns.

**Claims And Evidence:**

Yes

**Requested Changes:**

- Improve the REX repository.
- Please, clarify the error metric $v_\text{path}$ and $e_\text{path}$ used in Table 2.
- Address the minor points.

**Strengths And Weaknesses:**

**Strengths:**

- The experimental evaluation is convincing. Authors present results on two sim2real transfer tasks.
- The paper reads very well and feels polished. Figures are of high quality.
- The authors submitted their code to reproduce their experiments.

**Weaknesses:**

- My main point of criticism is the quality of the code repository. Since this paper revolves around an implementation, the code repository is a central component of the submission and should therefore be of high quality. The following points regarding the repository should be addressed before publication:
    1. The source code is mostly contained in two unstructured folders. I would suggest that the authors refactor their code and split it into two parts: (i) general code that can be used by the readers to solve their sim2real problems and (ii) code that is specific to the experiments presented by the authors (e.g., the contents of the notebooks folder or the pendulum folder).
    2. There is only a minimal documentation. I would expect a polished documentation of the functions and classes that future users can use as reference. I would suggest that the authors write a documentation using pydoc or similar documentation generator.

    3. I appreciate that there are some instructions that guide a potential user of the repository to solve their problem with the REX package, however those instructions should be more prominent within the repository/documentation.

    4. There are no tests. I suggest the authors write some unit tests (e.g. using pytest) that help maintaining the repository.

    5. Please follow the PEP 8 code style.

- Minor points:

    1. Code shown in Figure 3 (c) does not follow PEP8 code style. I would suggest to remove the unnecessary whitespaces.

    2. Computational complexity is not the right term when referring to a computational runtime analysis.

---

> ### Author Response · Authors · 2024-12-15
> **Response to Reviewer qxE9**
>
> Dear Reviewer qxE9,
>
> Thank you for your review. We have made the changes to the manuscript, which are highlighted in blue. Below, we address your concerns and suggestions that require more detailed responses.
>
> =====Quality of the Repository=====
>
> First off, due to the anonymous review process, we had to anonymize the repository, resulting in a slimmed-down version of the repository.
> In hindsight, you are correct that this does not allow for a thorough evaluation of the repository.
>
> That said, we have improved the repository since the last submission.
> Regarding the folder structure, the code within the "pendulum" directory was initially specific to the examples, while all general-purpose code was placed in the top-level folder.
> We believe that excessive directory nesting can clutter the repository and make navigation more challenging as long as the codebase is relatively small (as we believe it is in this case).
> However, we recognize the importance of clearly distinguishing between general code and example-specific code.
> To address this, we restructured the repository by introducing a new "examples" directory and moving the "pendulum" folder into it, anticipating future examples to be added.
> The updated repository structure is as follows:
>
> ```
> rex/                  # Project metadata
> rex/tests/          # Unit and integration tests
> rex/docs/            # Documentation files
> rex/examples/       # Notebooks that use example-specific modules.
> rex/rex/            # General-purpose code
> rex/rex/examples/ # Example-specific modules
> rex/rex/examples/pendulum/   # Pendulum-specific implementation
> ```
>
> Note that the example-specific modules are placed within the rex module to ensure they are shipped with the general code on PyPI.
> This allows seamless use in Google Colab with a simple `pip install rex`, avoiding the need for additional file uploads and potential namespace clashes if the examples were placed in a separate package.
>
> We have also added 99 unit tests that cover 97% of the codebase and an integration test based on the sim2real notebook.
> The code has been made fully compliant with PEP8 and Flake8 rules, and we have included pre-commit hooks for code formatting using Black.
> Instructions for contributing to the repository have also been added.
> Additionally, we have added documentation using mkdocs and plan to host it on GitHub Pages once anonymity is lifted and the repository can be made public.
> Note that some links in the docs are deliberately broken to prevent accidental disclosure of our identities.
> As per your suggestion, we have given the instructions for using REX a more prominent place in the documentation by incorporating them directly into the documentation itself.
> We intend to add more examples and tutorials to the documentation in the future, and we hope that the current documentation provides a sufficient starting point for users to get started with REX.
> Finally, we have added workflow files for CI/CD and PyPi deployment via GitHub Actions.
>
> =====Minor points=====
>
> - We have added definitions of the error metrics in the caption of Table 3.
> - As per your suggestion, Section 4.3 has been renamed to "Computational Runtime Analysis," and we have reformulated the text where necessary.
> - We have removed the whitespace in the caption of Figure 3c. However, we note that formatting the code example remains challenging due to the limited space in the column. We have made it as readable as possible but are open to further suggestions for improvement.
>
> Hopefully, these changes address your concerns about the quality of the repository and the minor points you raised.
> Thank you for your feedback, and we welcome any additional comments or questions you may have.

---

> > ### Comment · Reviewer_qxE9 · 2024-12-20
> >
> > Thank you for your response! All my concerns have been addressed.

---

### Review · Reviewer_6SbQ · 2024-11-21

**Summary Of Contributions:**

This paper presents a framework, REX, designed to bridge the gap between simulation and real-world robotic systems. The contributions and key aspects of this work can be summarized in the following 3 categories:

**Introduction of REX framework**
1. A graph-based simulation model optimized for accelerator hardware.
2. Explicit modeling of asynchronous dynamics, measurable and hidden delays (in communication, computation, sensing, and actuation), inherent in real-world robotic systems.

**Integration with advanced tools**
1. Leveraging JAX for parallelized computation, automatic differentiation, and high-performance training.
2. Compatibility with multiple physics engines (e.g., Brax, MuJoCo) within the graph-based model, allowing modular and extensible simulations.

**Experimental Validation**
1. Demonstrated the framework’s capabilities on two real-world systems: a pendulum swing-up task and a quadrotor path-following task.
2. The pendulum experiments highlighted the framework’s ability to estimate system dynamics and delays, achieving accurate sim2real transfer with delay compensation techniques.
3. The quadrotor experiments showcased scalability to higher-dimensional systems and emphasized the critical role of delay-aware training.

This paper's key contribution lies in its comprehensive approach to addressing latency and dynamics mismatch in sim2real systems. By combining robust delay modeling, system identification, and efficient parallelized computation, it aims to advance the sota in robotic learning and simulation frameworks.

**Audience:**

Yes

**Claims And Evidence:**

Yes

**Requested Changes:**

1. Including details about GMM modeling, and analysis on sensitivity to downstream performance. Analysis on minimum data requirements to train this model would be a plus. (critical)
2. Including details of different components included in the paper (dynamics models and optimization strategies used in the experiments). (critical)
3. Discussion around potential scaling-up to more complex real-world robotic systems. (strengthen)
4. Comparisons with existing frameworks like EAGERx and Drake, potentially a more detailed quantitative evaluation against these alternatives. (critical)

**Strengths And Weaknesses:**

**Strengths**

1. The introduction of a graph-based simulation framework (REX) that explicitly models latency, asynchronous dynamics, and hierarchical systems is a significant contribution. I am unable to judge the novelty of this approach. The framework successfully bridges the sim2real gap by addressing challenges in system delays due to communication, computation, sensing and actuation.
2. The use of Gaussian Mixture Models (GMMs) for stochastic delay modeling and Smith Predictor-based estimators for delay compensation is interesting.
3. The framework supports integration with multiple physics engines (e.g., Brax, MuJoCo), making it adaptable for various robotic systems.
4. Demonstrated effectiveness on two distinct real-world robotic systems: pendulum swing-up and quadrotor path following. Experimental results validate improvements in sim2real transfer and showcase the framework's ability to identify and compensate for delays accurately.
5. Effective use of hardware acceleration through JAX for both training and runtime execution.

**Weaknesses**

1. The paper does not lay out details about GMM modeling. How sensitive is downstream performance to the accuracy of this learned model? What are the minimum data requirements to train this model?
2. Modeling hidden delays requires specifying a maximum and minimum bound for each delay.
3. Usage of multiple out-of-the-box components, details of which are not included in the paper - e.g. the lightweight dynamics model (Dernet et al., 2020) and CMA-E Strategy used for parameter optimization. Having the required background details (e.g. in the appendix) makes the paper self-contained, which I believe is important in a good paper.
4. What are v_path and e_path in Table 2? Missing exact definitions of these error metrics.
5. While the experiments seem comprehensive, they are limited to controlled environments (pendulum and quadrotor tasks). Extending the validation to more diverse or complex real-world robotic systems could improve the generalizability of the framework.
6. The graph-based model’s scalability to extremely large or highly complex systems (e.g., multi-agent setups or industrial robotics) is not fully explored. Providing insights into potential bottlenecks and solutions for large-scale systems would be useful.
7. While comparisons with existing frameworks like EAGERx and Drake are mentioned, a more detailed quantitative evaluation against these alternatives in terms of performance, accuracy, and scalability would strengthen the claims.

---

> ### Author Response · Authors · 2024-12-15
> **Response to Reviewer 6SbQ**
>
> Dear Reviewer 6SbQ,
>
> Thank you for your review. We have made the changes to the manuscript, which are highlighted in blue. Below, we address your concerns and suggestions that require more detailed responses.
>
> ====GMM modeling====
>
> We have included an appendix detailing the GMM fitting procedure with measurable delays and a discussion of the data requirements. Typically, under a minute of data is sufficient, depending on the system's jitter level. Additionally, the accuracy of the fitted GMM model can be quickly assessed using the provided visualization tools. If certain modes are not captured, the number of components in the GMM can be increased to enhance the model's accuracy.
> Anecdotally, we have observed that the exact number of mixture components is not critical, provided that the model captures the key modes of delay distribution. While an ablation study on the sensitivity of downstream performance to the accuracy of delay modeling could be informative, such an analysis would be highly system-specific, as sensitivity depends on the magnitude of delays relative to the system's dominant time scales. We believe the current results sufficiently emphasize the importance of latency simulation and demonstrate the effectiveness of modeling delays as GMMs to bridge the sim-to-real gap.
>
> ====Min-max specification====
>
> Regarding the need to specify a minimum and maximum delay, we argue that this is a reasonable assumption, as most optimization algorithms require a defined search space. In the `sim2real.ipynb` notebook, we intentionally set the min-max range `(0, 0.30)` seconds to be 30 times larger than the actual delay of 0.01 seconds.
> Still, the optimization algorithm identified the correct delay values, showing robustness to min-max range variations.
> We have added a sentence in the limitations section to clarify this point.
>
> ====Detailing components====
>
> We added appendices detailing the UKF, the CMA-ES, and PPO.
> The appendices overview the framework's components, including hyperparameters, equations, and implementation, making the paper more self-contained.
>
> ====Scaling to larger & more complex systems====
>
> As systems grow in complexity, we believe our graph-based framework is well-suited to handle the increasing number of nodes and edges representing interactions within and between robots.
> For instance, a robot can be modeled as a subgraph within a larger system, enabling the representation of complex systems and asynchronous interactions.
> Efficiently parallelizing these simulations is critical, and we leverage [super2024], which demonstrates efficient compilation for systems with over 64 nodes and 256 edges.
>
> However, as systems scale to extremely large configurations, efficiently parallelizing full asynchronicity for every node can become a bottleneck.
> In such cases, the task dictates the level of detail required, and a graph-based framework provides the flexibility to scale up or down based on the user's needs.
> For instance, users may choose to simplify the task, for example, by focusing on interactions between robots, modeling entire robots as single nodes, in order to mitigate these challenges and maintain scalability in large-scale systems.
>
> The complexity of calculations within nodes is managed by our choice of JAX [jax2018], known for its versatility in handling advanced computations (e.g., Brax [brax2021] for physics simulations and [renderGH] for rendering).
> JAX also supports parallelization and distributed computing, enabling scalability to larger systems and multi-device setups.
>
> We have also added a dedicated paragraph in the limitations section of the paper to summarize how our framework handles scaling to larger and more complex systems, including potential bottlenecks and solutions to address them.
>
> ====Comparison with related frameworks====
>
> We have included a table (i.e., Tab. 1 in the paper) that qualitatively compares the features of related sim2real frameworks, aiming to help readers better understand the advantages of our framework over existing approaches.
>
> Additionally, we have added a runtime comparison of REX, Drake, and EAGERx, as these frameworks support the simulation of dynamical systems of comparable complexity that can be described by ordinary differential equations (ODEs). We have omitted Orbit from this comparison because it is tightly coupled with the PhysX simulator and does not support custom ODE-based simulations.
>
> We hope these changes address your concerns and suggestions. Thank you for your feedback, and we welcome any additional comments or questions you may have.
>
> - [jax2018] Compiling machine learning programs via high-level tracing. (Frostig et al., 2018 SysML)
> - [super2024] Efficient Parallelized Simulation of Cyber-Physical Systems. (van der Heijden et al. TMLR 2024)
> - [brax2021] Brax--a differentiable physics engine for large scale rigid body simulation. (Freeman et al. NeurIPS 2021)
> - [renderGH] jaxrender (https://github.com/JoeyTeng/jaxrenderer).

---

### Review · Reviewer_jGJv · 2024-12-02

**Summary Of Contributions:**

This paper contributes a framework/implementation to improve robotics simulation. In particular, the proposed framework contributes graph-based simulation model to more effectively model dynamics that come from communication and motor latency between various asynchronous systems in robotics. The goal is for this to lead to more effective sim2real policies. Additionally, the framework is built on Jax, Brax and aims to best utilise parallelzation and hardware accelerators.

**Audience:**

Yes

**Broader Impact Concerns:**

No major concerns.

**Claims And Evidence:**

Yes

**Requested Changes:**

No major changes. But if you could attach some video/picture evidence of real-world experiments conducted which better visualise the difference between a policy trained with REX vs not trained with REX and its siginicance in the real-world, that would be great.

**Strengths And Weaknesses:**

The paper brings up an important problem in robotics and sim2real frameworks. Paper is well written and the library/framework would be a nice contribution to the robotics/ML community. I do have some questions about the experimental evidence and comparisons as below:

Questions/Improvement Points:
- I am curious how this stacks up against other simulators which do not have this. Other simulators and briefly mentioned in the Sim2Real related work section which also support hardware acceleration but maybe does not model the latency dynamics of the different control loops/systems. For example, IsaacGym/Sim is commonly used now in many different papers and publications, especially in legged robotics where we have seen some really cool results. What is the comparison here? Why is it not important for them? It would be nice to discuss this.
- I think it would significantly improve and strengthen the paper to show pictures/videos of the real-world experiments conducted, along with its outcomes to allow readers to better visualise the benefits of the simulator.

---

> ### Author Response · Authors · 2024-12-15
> **Response to Reviewer jGJv**
>
> Dear Reviewer jGJv,
>
> Thank you for your review. We have made the changes to the manuscript, which are highlighted in blue. Below, we address your concerns and suggestions that require more detailed responses.
>
> =====Video=====
>
> We have added a supplementary video showcasing real-world sim2real experiments.
> The video highlights the impact of latency in both experiments.
> Capturing the quadrotor experiments was challenging due to the small size of the Crazyflie 2.1 and the camera setup we have in the lab.
> The video with low light conditions best represents the real-world experiments, and especially for the small `radius=0.5m` experiments of Fig. 9c, the difference between delay and no delay is clearly visible in the video.
>
> =====Importance of Latency Dynamics=====
>
> We believe that latency modeling was, in fact, crucial for sim2real transfer, as also evidenced by its impact on legged robotics.
> For instance, [rudin2022] had to modify IsaacGym to include a custom actuator model from [hwangbo2019], which accounts for control signal delays caused by hardware/software layers. Similarly, [tan2018] demonstrated that controller latency was a significant factor in the sim2real gap for quadruped robots. While frameworks like Orbit [mittal2023] incorporate actuator and sensor delays, they lack support for latency originating from other components, such as control loops. This gap is particularly critical for methods like Model Predictive Control (MPC), where latency compensation is essential, as highlighted by [yang2020], who emphasized the importance of compensating for planner latency in their legged robot deployment.
>
> Our framework unifies latency modeling across sensor, actuator, and computational delays, addressing critical challenges in sim2real transfer. It also incorporates tools for estimating and compensating for stochastic delays. The importance of modeling the randomness of delays is underscored by ongoing efforts in the Drake framework [drake2019], which seeks to model the "highly variable" computation times of planners due to both computational load and inherent runtime variability [drakeGH]. By offering comprehensive and principled latency modeling, our framework reduces the sim2real gap more effectively than ad-hoc approaches like aggressive domain randomization, which can diminish performance.
>
> In the previous version, we mentioned [yang2020] and [tan2018] in the Delay Compensation paragraph of the Related Works section. In this revision, we have additionally emphasized [rudin2022]'s use of a hybrid simulator [hwangbo2019] to address delays.
>
> We hope this clarifies the importance of latency dynamics in our framework and the broader context of sim2real transfer in legged robotics.
> Thank you for your feedback, and we welcome any additional comments or questions you may have.
>
> - [drakeGH] Idea: modeling long-running asynchronous computation in the system's framework (https://github.com/RobotLocomotion/drake/issues/11093#issue-427282915)
>
> - [mittal2023] Orbit: A Unified Simulation Framework for Interactive Robot Learning Environments. (Mittal et al. RA-L 2023)
>
> - [yang2020] Data efficient reinforcement learning for legged robots. (Yang et al. CoRL 2020)
>
> - [tan2018] Sim-to-Real: Learning Agile Locomotion For Quadruped Robots. (Tan et al. arXiv 2018)
>
> - [rudin2022] Learning to Walk in Minutes Using Massively Parallel Deep Reinforcement Learning. (Rudin et al. CoRL 2022)
>
> - [hwangbo2019] Learning agile and dynamic motor skills for legged robots. (Hwangbo et al. Science Robotics 2019)
>
> - [drake2019] Drake: Model-based design and verification for robotics. (Tedrake et al., 2019)

---

### Decision · Action_Editor_wHCY · 2025-01-21

**Recommendation:** Accept as is

**Comment:**

Reviewers are satisfied with the authors' responses and improvements over the original reviews and feedback, so I am recommending that the paper be accepted as is.

**Audience:**

This work will be of interest to robotics researchers, as well as other TMLR audiences that deal with the modelling of asynchronous systems.

**Claims And Evidence:**

This paper proposes a differentiable sim2real framework written in JAX that simulates latency and the effects it has across the components of robotics systems. The framework can simulate asynchronous, hierarchical systems that are frequently present in real robotic systems, and models multiple types of delays, including communication, actuation, and sensing. The paper also proposes effective compensation methods for dealing with all of these sources of delays. The framework enables dynamics estimation, system identification, as well as sim2real transfer. The framework is evaluated both on simulated and real systems.